# Tracking and Linking of Microparticle Trajectories During Mode-Coupling Induced Melting in a Two-Dimensional Complex Plasma Crystal

**DOI:** 10.3390/jimaging5030041

**Published:** 2019-03-16

**Authors:** Lénaïc Couëdel, Vladimir Nosenko

**Affiliations:** 1Department of Physics and Engineering Physics, University of Saskatchewan, Saskatoon, SK S7N 5E2, Canada; 2CNRS, Aix-Marseille Université, PIIM, UMR 7345, 13397 Marseille CEDEX 20, France; 3Institut für Materialphysik im Weltraum, Deutsches Zentrum für Luft- und Raumfahrt (DLR), D-82234 Weßling, Germany

**Keywords:** complex plasma crystals, particle tracking, mode coupling instability

## Abstract

In this article, a strategy to track microparticles and link their trajectories adapted to the study of the melting of a quasi two-dimensional complex plasma crystal induced by the mode-coupling instability is presented. Because of the three-dimensional nature of the microparticle motions and the inhomogeneities of the illuminating laser light sheet, the scattered light intensity can change significantly between two frames, making the detection of the microparticles and the linking of their trajectories quite challenging. Thanks to a two-pass noise removal process based on Gaussian blurring of the original frames using two different kernel widths, the signal-to-noise ratio was increased to a level that allowed a better intensity thresholding of different regions of the images and, therefore, the tracking of the poorly illuminated microparticles. Then, by predicting the positions of the microparticles based on their previous positions, long particle trajectories could be reconstructed, allowing accurate measurement of the evolution of the microparticle energies and the evolution of the monolayer properties.

## 1. Introduction

Complex (or dusty) plasmas are partially ionised gases containing microparticles. Due to the collection of the surrounding ions and electrons, the microparticles are (negatively) charged [1,2,3]. In laboratory experiments, injected monodisperse microspheres levitate at the same height above the confining electrode and form a monolayer [4,5]. Due to the interactions between the microspheres, the monolayer can under specific conditions crystallise (i.e., arrange itself into an ordered structure) and form the so-called two-dimensional (2D) complex plasma crystal [6,7,8,9,10,11].

In 2D complex plasma crystals, microparticles are easily observable by recording the scattered light of a horizontal laser sheet using a (high speed) video camera. It is therefore possible to obtain a rather complete picture about the state of the whole system of particles in the kinetic (x,v)-space. It offers a great advantage for the study of collective processes occurring in collective media. For this reason, 2D complex plasma crystals are often used to study generic phenomena happening in liquids and crystals at the particle (kinetic) level [3]. For example, they have been used to study melting and recrystallisation [12,13,14,15], mass and heat diffusion [16], solitons, and shocks [17].

Any strongly coupled system, such as 2D complex plasmas, supports two in-plane wave modes. In crystals, one mode is longitudinal and the other is transverse and they both have an acoustic dispersion [18]. Because the microparticles are confined by the sheath electric field, the strength of the vertical confinement is finite, allowing out-of-plane motion of the microparticles. Consequently, there is a third fundamental out-of-plane wave mode with negative optical dispersion [19,20]. The wave dispersion depends directly on the 2D complex plasma crystal parameters allowing one to use the propagation of dust-lattice (DL) waves as a diagnostic to determine these parameters [8,21,22,23].

Since in most experiments, the microparticles levitate in the sheath of a capacitively coupled radio frequency (rf) discharge, the ion flow coming from the bulk plasma and directed toward the electrode is focused downstream of each negatively charged microparticle of the monolayer. A perturbed region, known as the “ion (or plasma) wake”, is then created below each microparticle and exerts an attractive force on the neighbouring particles. Ion wakes act as an (external) “third body” in the interparticle interaction, resulting in nonreciprocal particle pair interactions [24,25,26,27]. It is known that under specific conditions, it will lead to the formation of an unstable hybrid mode (when the out-of-plane mode crosses the in-plane longitudinal mode), which can trigger the rapid melting of the 2D complex plasma crystal: The mode coupling instability (MCI) [28,29,30,31,32]. The hybrid mode has clear fingerprints: Critical angular dependence, a mixed polarisation, distinct thresholds [30], and synchronisation of the particle motion [33]. Wake-induced mode coupling is also possible in liquid complex plasma monolayers [31]. In this case, the confinement and dust particle density thresholds disappear. Moreover, the instability has a higher growth rate than in the crystalline phase. It is therefore possible to trigger sporadic melting of a stable crystal, which is not too far from the crystalline MCI threshold, by applying a sufficiently strong mechanical perturbation. The full melting of a two-dimensional plasma crystal was induced in a principally stable monolayer by localised laser stimulation [34]. MCI-induced melting has been used to study generic phenomena such as thermoacoustic instability [35] and combustion in 2D media [36].

The study of collective motion in complex plasmas requires high-speed imaging of the microparticle system [37]. For (quasi-) 2D complex plasmas, a laser light sheet is usually used to illuminate the monolayer of particles, and the scattered light is then recorded by the camera to monitor dust trajectories on the plane. Megapixel cameras or higher resolution cameras coupled to good close up lenses are used. The achieved resolution is generally of a few tens of micrometres per pixel. The frame rate of the camera should also be chosen properly. In 2D complex plasma crystals, the maximum frequency of the different wave modes is generally around 10–40 Hz [21,30,38]. Consequently, according to the Nyquist criterion, the sampling frequency should be 20–80 Hz. In practice, a much higher frame rate of a few hundred frames per second is usually used. Consequently, after each experiment, large video files containing a few thousand frames need to be processed. Each frame usually contains a great number of microparticles. After efficient analysis, statistics of some important physical quantities such as the dust kinetic temperature can be extracted. Two common analysis techniques are particle image velocimetry (PIV) and particle tracking velocimetry (PTV). In PIV, the frames are split into a large number of interrogation windows containing a few particles. Displacement and velocity vectors can then be calculated for each window using autocorrelation or cross-correlation techniques [39]. In PTV, individual particles are tracked through as many consecutive frames as possible, allowing the reconstruction of many detailed trajectories and the reconstruction of the velocity distribution function with a great accuracy. PIV has been employed successfully to extract velocity fields to study the propagation of waves in complex plasmas [40,41,42]. Since PIV does not allow the identification of single particle trajectories, it is therefore not adapted to study phenomena, such as dislocations, for which PTV is more adapted [43,44,45]. In the particular case of MCI, even though PIV has been successfully employed to track the melting front [42], the preferred technique is PTV, which allows very detailed studies of the melting kinetics at the different stages of the instability [30,33,34,36,46,47,48].

PTV techniques used in complex plasma studies are generally similar to those used for colloids and biological systems (such as the one presented in References [49,50,51,52]. The two main steps of PTV are: (i) Locating the particles in each frame and (ii) linking the locations into trajectories. Both steps are very sensitive to the quality of the recorded video. During step (i), the main challenge is to distinguish the particle from the surrounding image background. Here, the signal-to-noise ratio (SNR) plays a crucial role. Many approaches to particle detection exist in the literature: Simple thresholding, detection of local maxima, filtering such as Gaussian, Laplacian of Gaussian, difference of Gaussian, etc. [51]. During step (ii), the linking approaches can vary from very simple nearest neighbour association to multiframe linking with trajectory prediction. Many studies have been devoted to the optimisation of the tracking and trajectory linking algorithms in dusty plasmas, including error analysis and advanced diagnostics for trajectory linking (see, for example, [53,54,55,56,57]) and many software packages are readily available for particle tracking [58]. Many attempts have been made to find the best tracking–linking strategy, especially for the study of biological systems [51]. However, as stated by Chenouard et al. [51], “there exists no universally best method for particle tracking” and “a method reported to work for certain experiments may not be the right choice for [another] application”.

In this article, we present a tracking and linking strategy adapted to the study of MCI and MCI-induced 2D complex plasma crystal melting. Indeed, in this case, the challenge arises from the detection and tracking of the particles in the melted area. The reasons for the difficulties are manifold: (i) If the frame rate is too low, dust particles can appear as streaks making the positioning inaccurate. This problem can be resolved by increasing the frame rate (which results in a decreased exposure time). (ii) If the particles move too much between two consecutive frames, it becomes very complicated to identify their trajectories during image post analysis, since the trajectories of neighbouring particles can cross. During MCI-induced melting, the microparticles acquire relatively large velocities (especially in the melted region), making the linking of the trajectories extremely challenging. (iii) The particle motion is not two-dimensional, and the out-of-plane component in the melted region is much more important than in the crystalline region. Consequently, the microparticles tend to leave the illumination plane of the laser sheet and become virtually undetectable. Since the laser light sheet has a Gaussian profile, it is therefore necessary to develop an algorithm that locates the particles while keeping the detection of false particles (due to pixel noise) as low as possible. Note that the aim of this study is not to present the best method for particle tracking in quasi-2D complex plasma crystals, but rather to present an adapted method that gives results of proper quality for the detailed investigation of the melting of such systems. The article is structured as follows: Section 2 presents the experimental setup, Section 3 presents the chosen tracking and linking strategy: Thanks to a two-pass noise removal process based on Gaussian blurring of the original frames using two different kernel widths, the signal-to-noise ratio was increased to a level that allowed a better intensity thresholding of different regions of the images and, therefore, the tracking of the poorly illuminated microparticles. Then, by predicting the positions of the microparticles based on their previous positions, long particle trajectories could be reconstructed, allowing accurate measurement of the evolution of the microparticle energies and the evolution of the monolayer properties. The position prediction method is compared to the quasi-static approximation strategy commonly used for trajectory linking. The method is applied to MCI-induced melting of a 2D complex plasma crystal. Finally, Section 4 recalls the main results and concludes.

## 2. Experimental Setup

The experiment was performed in a modified GEC chamber. A capacitively coupled rf glow discharge was sustained at 13.56 MHz. The argon pressure could be varied between 0.4 Pa and 2 Pa. The forward rf power was between 5 W and 20 W. The plasma parameters in the bulk discharge were measured using a Langmuir probe. The electron temperature was $T_{e}∼2.5$ eV, and the electron density was $n_{e}∼2\times 10^{9}$ cm${}^{-3}$ at a pressure p=0.66 Pa and a forward rf power $P_{W}=20$ W [59]. A schematic of the experimental setup is presented in Figure 1a. By levitating calibrated melamine–formaldehyde spherical particles, a horizontal monolayer was formed in the plasma sheath above the lower rf electrode. The particles had a diameter of 9.19±0.09μm. The particle monolayer was illuminated using two laser sheets: A vertical one and a horizontal one. A Photron FASTCAM 1024 PCI camera (Photron, Tokyo, Japan) (1024×1024 pixels) at a speed of 250 frames per second was used to image the particles through the window at the top of the vacuum vessel (see Figure 1b). The resolution of this top view camera was 41.8 μm/pixel. The frame rate used was high enough that when comparing two consecutive frames, the microparticle motion was hardly noticeable in the crystalline region and limited to a maximum of a few pixels in the melted region.

An additional side-view camera (Basler Ace ACA640-100GM (Basler AG, Ahrensburg, Germany) at 103.56 fps) was used to check that no extra particles were levitating below or above the main layer (perfect monolayer, see Figure 1c). Depending on the number of injected particles, the diameter of the obtained crystalline structure was up to ∼60 mm.

In the present experiment, a crystalline monolayer with the right parameters to trigger the mode-coupling instability [30,33,46] was created to study its rapid melting [34,35,36]. For this purpose, the microparticles were injected at a pressure p=0.6 Pa and rf power $P_{W}=20$ W. Then, the pressure was increased up to p=1.6 Pa and the rf power reduced as much as possible in order to keep only monodisperse microparticles and drop all the agglomerates levitating below the main layer. The power was then restored back to $P_{W}=20$ W and the pressure reduced to p=1.2 Pa. With these discharge parameters, the mean number density of the crystalline monolayer in the camera field of view was ∼530 cm${}^{-2}$ (∼9800 particles in each frame). The mode-coupling instability was then triggered by gradually reducing the rf power and started at $P_{W}=14$ W. The particle motion was recorded for 9.6 s (2400 frames).

## 3. Data Analysis

The analysis of the experimental video is divided in two main parts: (i) Locating the particles in each frame and (ii) linking the particle locations into individual particle trajectories.

### 3.1. Particle Location

In a given frame, the local brightness maxima are usually candidates for particle locations. It is usual practice to binarise the original image: Pixels k having intensities $I_{k}$ above a threshold value $I_{\mathrm{th}}$ are set to a value of $B_{k}=$1 (bright pixels) in the binary image *D*, and the other pixels are set to 0. Then, each group of contiguous bright pixels in *D* (blobs) can be used to compute the particle location by calculating the centre of mass of the blob:(1)$$R_{\mathrm{blob}}=\frac{\sum_{k}R_{k}I_{k}}{\sum_{k}I_{k}},$$
where $R_{\mathrm{blob}}$ is the position of the blob and $R_{k}$ the position of the blob pixel k of intensity $I_{k}$ in the original image.

However, direct thresholding of the raw frames is in practice not feasible, since noise can render the process of identification of the pixel blobs corresponding to microparticles very inefficient. Two types of noise usually coexist in the frame: (i) Noise due to inhomogeneous illumination and (ii) random noise due to the digitisation of the image in the video camera. In the presented experiment, it resulted in an SNR of ∼10 at the center of the frame and ∼3 at the edges, where $\mathrm{SNR}=\left(I_{p}-I_{b}\right)/\sqrt{I_{p}}$, with $I_{p}$ being the peak particle intensity and $I_{b}$ the mean background intensity. For example, in Figure 2, binary images obtained from direct thresholding of a raw frame (Figure 2a) are presented. In Figure 2b, the threshold value was fixed to $I_{\mathrm{th}}=3$. As can be seen, at the edge of the picture (zoom in Figure 2c), the threshold value seems to be adequate and only the bright pixels corresponding to the particles were kept. However, in the centre of the frame (zoom in Figure 2d), one can see that many pixels corresponding to background noise were mistaken for microparticles. Obviously, this binarised image cannot be used for blob identification and location, since many fake particles would be detected and blobs corresponding to real microparticles could be deformed. In Figure 2e, the threshold value was fixed to $I_{\mathrm{th}}=6$. As can be seen, at the edge of the picture (zoom in Figure 2f), the threshold value is too high, and many low intensity pixels corresponding to the microparticles were discarded. However, in the centre of the frame (zoom in Figure 2g), background noise was now properly removed from the particle signal. Nevertheless, this binarised image cannot be used either for blob identification and location, since many microparticles remain now mistakenly undetected.

It is, however, possible to optimise blob detection by removing the noise induced by the imperfection of our video acquisition system. The background noise due to the inhomogeneities of the illumination system results in long-wavelength variation of the image background. The background variations can be seen in Figure 2a. They are even better evidenced in Figure 3a,h, in which the grey scale has been further compressed. This varying background is responsible for the failure when using a single-intensity threshold value to binarise the picture. Another source of noise is, as previously said, the random noise due to the digitisation in the video camera. It usually has a correlation length of $\lambda_{n}≃$ 1–2 pixels, and its amplitude does not usually exceed a few grey levels. The camera used in this experiment has a 10-bit pixel depth, out of which 8 bits were used, allowing to code the pixel intensities from 0 to 255. Intensity variations due to the digitisation noise were usually ΔI∼±1. This noise did not have a big influence when dealing with large bright spots but could introduce significant errors when the features to detect were not very bright and/or when they were defined only by a small number of pixels.

In our experiment, the inhomogeneities of the illumination system were significant over the length $\lambda_{\mathrm{BG}}≃4\;\mathrm{mm}≃100$ pixels. By convolving the original image (firstly converted to a matrix of floating numbers instead of unsigned 8-bit integers) with a Gaussian surface of revolution with a standard deviation $\sigma_{BG}=\lambda_{\mathrm{BG}}/2$, all the pixel blobs associated with the microparticles (which have typical areas of 2–6 pixels) were blurred away, and a “background image” $B_{1}$ that retained the background noise due to the inhomogeneities of the illumination system was created (the *imgaussfilt*
MATLAB^®^ function was used to compute *B*):(2)$$B_{1}\left(x,y\right)=\frac{\sum_{i,j=-w}^{w}A\left(x+i,y+j\right)\exp\left(-\frac{i^{2}+j^{2}}{2\sigma_{BG}^{2}}\right)}{\sum_{i,j=-w}^{w}\exp\left(-\frac{i^{2}+j^{2}}{2\sigma_{BG}^{2}}\right)},$$
where (x,y) are the indices of a pixel in the original image *A* and the computed background image $B_{1}$, and $w=\sigma_{BG}$ in pixels. A first filtered image is then calculated as (if $A_{{\mathrm{filt}}_{1}}\left(x,y\right)<0$, then it is set to 0):(3)$$A_{{\mathrm{filt}}_{1}}=A-B_{1}.$$

In Figure 3b,i, the results of the removal of long-wavelength background are shown in the centre and at the top right corner of the frame, respectively. As can be seen, the background noise has considerably decreased. This is especially true in the centre of the image (Figure 3b). However, there is still a bit of noise (especially at the edges) that could make the identification of the pixel blobs corresponding to the microparticles problematic. For instance, in Figure 3i, a lot of “salt and pepper” random noise is still visible, and the intensities of the noise pixels are barely lower than the “true” particle signal.

A second background image is then obtained by convolving $A_{{\mathrm{filt}}_{1}}$ with a Gaussian surface of revolution with standard deviation $\sigma_{RN}=\Delta$, where Δ is the mean interparticle distance in the experiment (in our case, Δ≃ 10 pixels). The pixel blobs associated with the microparticles are blurred away, and the “background image” $B_{2}$ retains only the intensity variations which correspond to the average noise intensity on the interparticle distance scale:(4)$$B_{2}\left(x,y\right)=\frac{\sum_{i,j=-w_{2}}^{w_{2}}A_{{\mathrm{filt}}_{1}}\left(x+i,y+j\right)\exp\left(-\frac{i^{2}+j^{2}}{2\sigma_{RN}^{2}}\right)}{\sum_{i,j=-w_{2}}^{w_{2}}\exp\left(-\frac{i^{2}+j^{2}}{2\sigma_{RN}^{2}}\right)},$$
where $w_{2}=\sigma_{RN}$ in pixels. Note that the blobs corresponding to the microparticles contributed slightly to $B_{2}$, but since $\sigma_{RN}$ is larger than them, this had a minimal effect. A new filtered image is then calculated (if $A_{{\mathrm{filt}}_{2}}\left(x,y\right)<0$, then it is set to 0):(5)$$A_{{\mathrm{filt}}_{2}}=A_{{\mathrm{filt}}_{1}}-B_{2}.$$

In Figure 3c,j, the results of the removal of $B_{2}$ are shown in the centre and at the top right corner of the frame, respectively. As can be seen, the noise level has further decreased. In the centre of the frame (Figure 3c), there is almost no remaining noise. At the edge (Figure 3j), the noise level is now manageable, and the frame is suitable for thresholding and creating the binary image.

One could argue that applying the $B_{2}$ filter to the original image *A* twice in a row would provide the same results, since $B_{2}$ is roughly the background intensity at the interparticle distance scale. This procedure was applied in Figure 3e,f in the centre of the frame and in Figure 3l,m at the top right corner of the frame. As can be seen, applying $B_{2}$ once gave results quite similar to $B_{1}$ (Figure 3e,l). However, repeating $B_{2}$ did not further improve the images (Figure 3f,m). Therefore, this procedure was not used in our analysis.

To obtain the particle locations, a general threshold $I_{\mathrm{th}}$ (in this experiment, $I_{\mathrm{th}}=1.8$) was then used to binarise the final filtered image $A_{{\mathrm{filt}}_{2}}$ (Figure 3d,k). As can be seen, now the remaining blobs corresponded to microparticles. Then, as described previously, each group of contiguous bright pixels (blobs) was used to compute the particle location with sub-pixel accuracy by calculating the centre of mass of the blob using the pixel intensity $I_{{\mathrm{filt}}_{2}}$ of $A_{{\mathrm{filt}}_{2}}$:(6)$$R_{\mathrm{blob}}=\frac{\sum_{k}R_{k}I_{{{\mathrm{filt}}_{2}}_{k}}}{\sum_{k}I_{{{\mathrm{filt}}_{2}}_{k}}}.$$

Since the blobs of pixels corresponding to the particles are quite small in the current experiment (3–5 pixels in the centre of the frame and 1–3 pixels at the edges), pixel locking of the particle positions is present and is particularly strong for the particles located at the edges of the video frames. Consequently, quantities relying on very accurate identification of the particle positions such as kinetic temperature in the crystalline phase cannot be accurately calculated. Note that it may be possible to slightly refine the positions of the particles to a greater accuracy by using a more advanced algorithm based, for example, on the fitting of each blob intensity with a 2D Gaussian surface (see, for example, References [50,52]). Nevertheless, the best way to suppress pixel locking is to increase the number of pixels per particle image, e.g., by using a higher resolution imaging system.

### 3.2. Trajectories (Track Linking)

After the microparticle positions have been recovered, one needs to link them from frame to frame in order to recover the trajectories of individual particles. To achieve this, for each particle found in frame N, one tries to find a corresponding particle in frame N+1. In practice, if two particles found in two consecutive frames are close to each other, they are most probably the same particle. The simplest algorithm originally created by Crocker and Grier [49] supposes a Brownian motion of the particles, meaning that there is no correlation in a particle velocity from frame N to frame N+1. Moreover, the frame acquisition rate being relatively high, the displacement of a microparticle between two frames is relatively small. Under these assumptions, for a particle *i* at position $R_{i}\left(t_{N}\right)$ in frame N, the most probable location $R_{i_{MP}}\left(t_{N+1}\right)$ is (quasi-static approximation):(7)$$R_{i_{MP}}\left(t_{N+1}\right)=R_{i}\left(t_{N}\right).$$

Consequently, for each particle *i* found in frame N, we look for the closest microparticle in a small region of radius $r_{\mathrm{search}}$ centred around $R_{i}\left(t_{N}\right)$. In order to limit mistracking (association of a wrong particle *j* in frame N+1 with particle *i* in frame N), $r_{\mathrm{search}}$ is limited to the mean particle radius (here, the radius of the blob of pixels is considered, not the physical radius of the particle) derived from the average pixel number per particle obtained from the particle location [50]. In this experiment, $r_{\mathrm{search}}=1.5$ pixels. This linking strategy was successfully applied to study wave modes in 2D complex plasma crystals [20] as well as the early studies of MCI [30,42,46].

However, when the MCI sets in, microparticles acquire large kinetic energies [30,36,42]. Even though video acquisition is usually done at a relatively large frame rate (250 fps in the presented experiment), microparticles can have the time to move significantly from one frame to the next. Under this condition, the quasi-static approximation fails if a particle moves more than the microparticle mean radius and its trajectory is discontinued. In order to avoid this, one can try to predict more accurately where the microparticles might be located in frame N+1 by using the information gathered from frames N and N-1. Indeed, by knowing the position of a particle *i* in frames N and N-1, it is possible to have a quick estimate of its velocity $v_{i}\left(t_{N}\right)$:(8)$$v_{i}\left(t_{N}\right)=\left(R_{i}\left(t_{N}\right)-R_{i}\left(t_{N-1}\right)\right)/\tau ,$$
where τ is the inverse frame rate. With this estimated velocity, it is then possible to predict the position $R_{i_{Pred}}\left(t_{N+1}\right)$ of the microparticle in frame N+1 (position prediction):(9)$$R_{i_{Pred}}\left(t_{N+1}\right)=R_{i}\left(t_{N}\right)+v_{i}\left(t_{N}\right)\tau .$$

To link the trajectories, we look for a particle in a small region of radius $r_{\mathrm{search}}$ centred around $R_{i_{Pred}}\left(t_{N+1}\right)$. In order to limit mistracking, $r_{\mathrm{search}}$ is now limited to twice the mean particle radius derived from the average pixel number per particle obtained from the particle location (in our case, $r_{\mathrm{search}}=3.0$ pixels). It is slightly increased compared to the previous method to allow for small microparticle velocity variations. However, in order for this method to work properly, one has to make a few hypotheses:At the start of the movie (i.e., frame 1), we assume that the velocity of each microparticle is 0. This is equivalent to the quasi-static approximation. For MCI-induced melting studies, the video usually starts when the monolayer is still in the crystalline phase, and it is therefore not a bad approximation.If a new particle appears in frame N+1 that was not in frame N, then the newly-appearing particle borrows the velocity of the closest particle already present in frame N.

After the microparticles were tracked (trajectory linking was performed using the Trackpy library [60]), the trajectories were further filtered. Indeed, even though noise has been reduced to a low manageable level before microparticle location, some spurious blobs might still be wrongly identified as real microparticles. These features were random, and they were usually not present in more than a few consecutive frames. Consequently, in order to eliminate these fake particles, the tracking algorithm was required to keep only microparticles that could be tracked for a minimum of $N_{\min}$ frames in a row. In our case, $N_{\min}=3$ frames.

### 3.3. Comparison of the Tracking Methods, Evolution of the Particle Monolayer

In Figure 4, the results of particle tracking using the quasi-static approximation (Figure 4a) and the position prediction (Figure 4b) at t=8.0 s (after the crystal has melted in its centre) are presented. As can be seen, using the quasi-static approximation resulted in losing many particles. The quasi-static approximation algorithm was not able to link the trajectories for $N_{\min}=3$ frames for a relatively large number of particles. Moreover, the number of new (rediscovered) particles (circled in red) was quite high, which can have an impact on the accuracy of the computed velocities and kinetic energies. By contrast, when predicting the position for trajectory linking, most of the particles were properly detected and linked.

With the particle trajectories, the velocity of the microparticles could be calculated. For this purpose, a Savitzky–Golay filter using a second order polynomial on 3-point window was applied (resulting in a piecewise exact fit of the trajectories). Since in each particle trajectory the data points are equally spaced (by the inverse frame rate), a single set of coefficients can be applied to each piece of a particle trajectory (the 3-point window) to calculate the first time derivative of the trajectory (the velocity) at the central point of the window. Thus, the microparticle velocities $v_{i}$ were obtained. Since the computed velocities at the beginning and the end of a trajectory were not as precise (due to the absence of data from the frame before or after), longer trajectories were better to recover more accurate velocity fields. Using the microparticle velocities, the 2D kinetic temperature $T_{\mathrm{kin}}\left(t_{N}\right)$ can be calculated:(10)$$k_{B}T_{\mathrm{kin}}\left(t_{N}\right)=\frac{1}{M}\sum_{i=1}^{M}\frac{1}{2}m_{d}{\left(v_{i}\left(t_{N}\right)-\langle v\left(t_{N}\right)\rangle \right)}^{2},$$
where *M* is the number of microparticles in frame N, $k_{B}$ is the Boltzmann constant, $m_{d}$ is the mass of a microparticle, and $\langle v\left(t_{N}\right)\rangle$ is the average velocity of the microparticles in frame N.

In Figure 5, the evolution of the microparticle kinetic temperature $T_{\mathrm{kin}}$ is shown using the two different linking strategies. As can be seen, while the kinetic temperature of the monolayer is not too high ($T_{\mathrm{kin}}\leq 20$ eV), both strategies give equivalent results. For instance, the exponential growth of the kinetic temperature due to MCI in the crystalline phase (from t∼4 s to t∼7 s) could be well identified in both cases. However, when the crystal melted and MCI entered in the fluid phase, the results were quite different: In the case of the quasi-static approximation, the energy growth rate barely increased, and the kinetic temperature saturated at ≳100 eV. This is due to the fact that fast particles moving more than 1.5 pixels per frame (corresponding to a kinetic energy of ∼470 eV) were lost. At t∼8.2 s, the energy growth rate decreased due to MCI saturation [34,36,61]. However, the measured kinetic temperature is underestimated because of the lost particle tracks. This is especially true for the quasi-static approximation method. Note that the relatively high kinetic temperature and quite large fluctuations at the beginning of the experiment (t≤ 4 s) were due to small mechanical vibrations of the experimental setup as well as pixel locking and digitisation noise [42,54].

In Figure 6, a detailed analysis of the crystal evolution during the MCI-induced melting is shown. All the parameters were obtained using the trajectories recovered using position prediction. In the central part of the crystal (delimited by the white square in Figure 6b), the microparticle pair correlation function g(r) was calculated. The position of its first maximum gave us the mean interparticle distance Δ=430±20
μm, which corresponded to a crystal number density ρ≃620 cm${}^{-2}$. The tracking data were also used to compute the particle velocity fluctuation spectra as follows. First, the particle current components $V_{s}\left(k,t\right)$ were calculated in the direction of interest *s*:(11)$$V_{s}\left(k,t\right)=\sum_{j=1}^{N}v_{s,j}\left(t\right)e^{-ık\cdot s_{j}\left(t\right)},$$
where $k=\{k_{x},k_{y}\}$ is a wave vector located in the horizontal plane. Here, *ı* is the imaginary unit, *j* is the particle index, $v_{s,j}\left(t\right)$ is the *s*-projection of the *j*-th particle velocity, $s_{j}=\left\{x_{j},y_{j}\right\}$ is its position, and *N* is the number of microparticles. Note that the x,y-axes were chosen as shown in Figure 1, allowing us to easily choose the direction of propagation of the waves with respect to the lattice principal axes. A fast Fourier transform in the time domain was then implemented to obtain the current fluctuation spectra. Finally, from the microparticle positions, Voronoi diagrams were constructed, and the value of the local bond-orientational order parameter $\psi_{6}$ was computed:(12)$$|\psi_{6}\left(r_{m}\right)\left|=\right|\frac{1}{N_{b}}\sum_{n=1}^{N_{b}}\exp\left(6i\theta_{mn}\right)|,$$
where $N_{b}$ denotes the number of nearest neighbours of the $m^{\mathrm{th}}$ particle at position $r_{m}$ and $\theta_{mn}$ is the angle between the bond of particles *m* and *n* and an arbitrary but fixed reference axis. $|\psi_{6}|=1$ indicates a perfectly hexagonal Voronoi cell. A decrease of $|\psi_{6}|$ indicates a deformation of the cell (crystal disturbance). It is a local diagnostic which is highly sensitive to any strain inside the crystalline structure. A change in $|\psi_{6}|$ is easily detectable and occurs before the appearance of defects (5-fold and 7-fold defects in 2D crystals).

In Figure 6a, the in-plane current fluctuation spectra obtained using the trajectories from t=1.6 s to t=5.6 s are plotted for a propagation angle $\theta =0^{\circ }$ compared to the crystal principal axis (see inset in Figure 1a). The long-wavelength part kΔ≤1 of the spectra can then be fitted by the theoretical dispersion relations [29]. Assuming pure Yukawa interactions, the microparticle charge number was estimated $Z_{d}=-18,000\pm 3500$ and the screening parameter $\kappa =\Delta /\lambda_{D}=1.10\pm 0.15$, where $\lambda_{D}$ is the Debye (screening) length. Note that due to MCI, a hot spot can be seen at the border of the first Brillouin zone as well as traces of mixed polarisation (traces of the out-of-plane (vertical) mode in the in-plane longitudinal mode [30]).

In Figure 6b–d, the maps of the monolayer showing the crystal defects, the local value of $|\psi_{6}|$, the local number density, and the local kinetic energy are shown. At the beginning of the instability (Figure 6b), the crystalline structure at the centre of the monolayer was almost perfect with very few defects. However, due to MCI, at the centre of the monolayer where it is at its densest, the kinetic energy was slightly higher and $|\psi_{6}|$ was slightly below 1 due to local deformation owing to the synchronised motion of the microparticles [33,47]. This effect could be also seen in the local particle number density with the appearance of fringes at the centre of the monolayer. At t=6 s (Figure 6c), the kinetic energy at the centre of the monolayer has grown high enough to partially destroy the crystalline order and lead to the appearance of defects. $|\psi_{6}|$ in this region is well below 1, and the number density fluctuations due to the monolayer entering the fluid regime were well pronounced. At t=8 s (Figure 6c), the monolayer was well into the fluid MCI regime [31]. The kinetic energy at the centre of the melted monolayer was very high, and the melted zone was expanding outwards. Many defects are visible. $|\psi_{6}|$ revealed the very sharp border of the melted zone. Large number density fluctuations were seen in the centre. Note that they were larger than the real fluctuations due to the 3D nature of the microparticle motion, which could not be tracked at all times (the results would be even worse using the quasi-static approximation). Dust density waves (“sound waves”) were also generated by the melted spot and observed in the number density and kinetic energy maps. In recent studies, MCI-induced melting was used to simulate rapid melting and flame front propagation in 2D matter [34,36] as well as thermoacoustic instability [35]. These studies would not have been possible without efficient tracking of the trajectories in all regions of the microparticle monolayer.

## 4. Conclusions

In this article, a strategy to track microparticles and link their trajectories adapted to the study of the melting of a quasi two-dimensional complex plasma crystal induced by MCI was presented. Due to the three-dimensional nature of the motion of the microparticles and because of the inhomogeneities of the illuminating laser light sheet, the intensity of the light scattered by a microparticle could change significantly between two frames during MCI-induced melting resulting in a broad range of SNR in a single frame and making the tracking of the microparticles and the linking of their trajectories quite challenging. Thanks to a two-pass noise removal process based on Gaussian blurring of the original frames using two different kernel widths, the signal-to-noise ratio was increased to a level that allowed a proper identification and location of the microparticles, including those with a low scattered light signal. Then, by predicting the positions of the microparticles based on their previous positions, long particle trajectories were reconstructed, allowing accurate measurement of the evolution of the microparticle energies and the evolution of the monolayer properties to the accuracy that cannot be matched by the algorithm assuming quasi-static microparticles. For example, because the fast microparticles could be tracked more efficiently, the maximum measured energy was higher, and the transition to fluid MCI could be better identified, allowing detailed investigation of the monolayer melting dynamics.

In future studies, the influence of the frame rate and pixel locking will be investigated to obtain cleaner information about the monolayer, especially in the crystalline phase [42,54]. A more accurate positioning algorithm based on the Gaussian fit of the luminosity blobs corresponding to the microparticles will be implemented [50,52]. Finally, a more powerful tracking algorithm based on the Kalman filter [56] will be implemented to study the fluid phase with an even greater accuracy.

## Figures and Tables

**Figure 1 jimaging-05-00041-f001:**
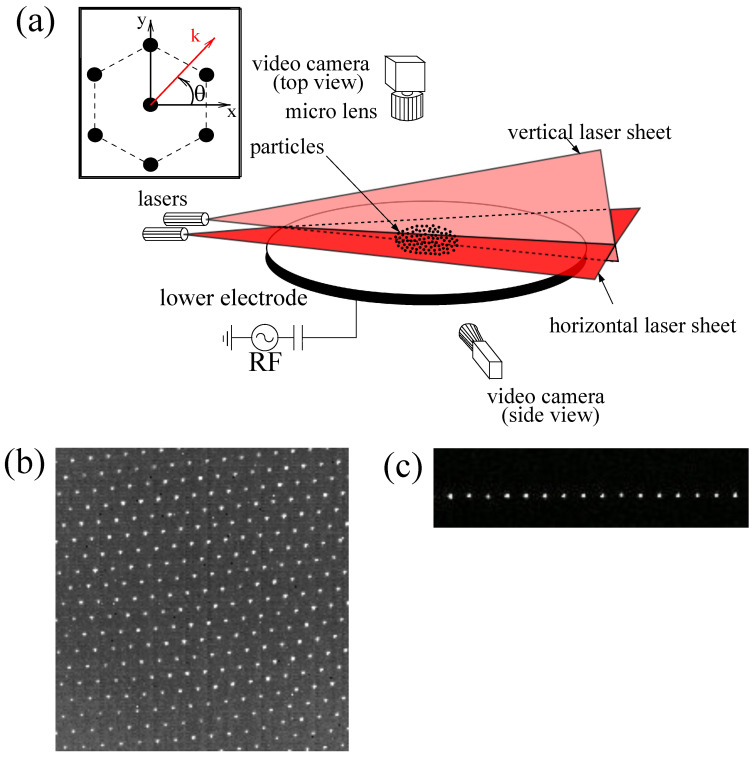
(**a**) Schematic diagram of the experimental setup. The inset shows the chosen orientation of the crystal lattice. (**b**) Top-view image of a 2D complex plasma crystal. (**c**) Side-view image of a 2D complex plasma crystal. Note that images (**b**,**c**) have been heavily reworked (nonlinear contrast and luminosity enhancements, Gaussian smoothing) and are therefore for illustration only.

**Figure 2 jimaging-05-00041-f002:**
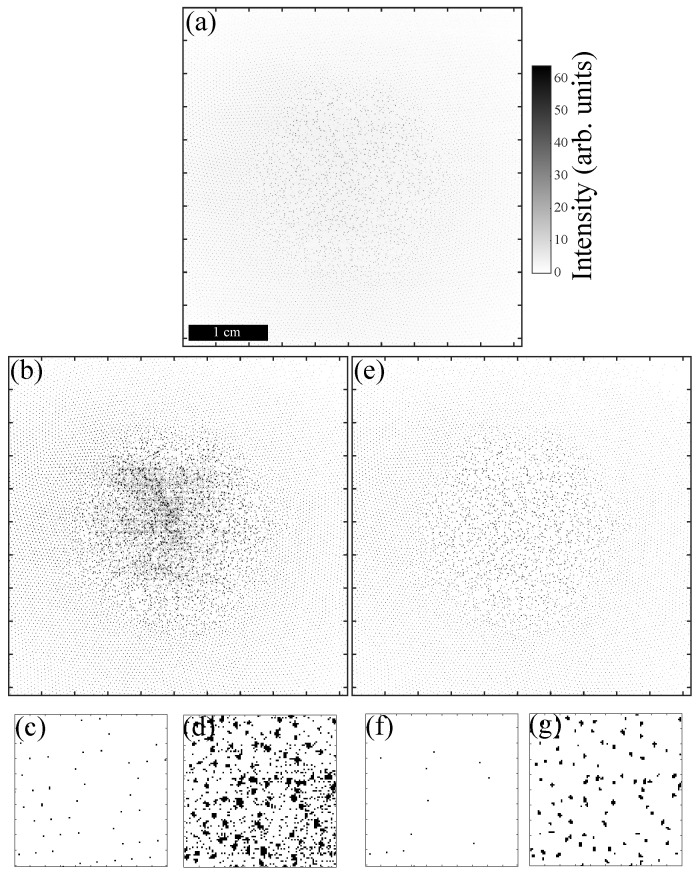
(**a**) Original frame (the intensity map was inverted for clarity); (**b**) Binary image obtained with a threshold of 3; (**c**) zoom of (**b**) at the top left corner of the frame; (**d**) zoom of (**b**) in the centre of the frame; (**e**) Binary image obtained with a threshold of 6; (**f**) zoom of (**e**) at the top left corner of the frame; (**g**) zoom of (**e**) in the centre of the frame. Note that the grey scale of the original image has been changed from 0–255 to 0–64 for better visibility (pixels artificially saturate if their intensity is greater than 64).

**Figure 3 jimaging-05-00041-f003:**
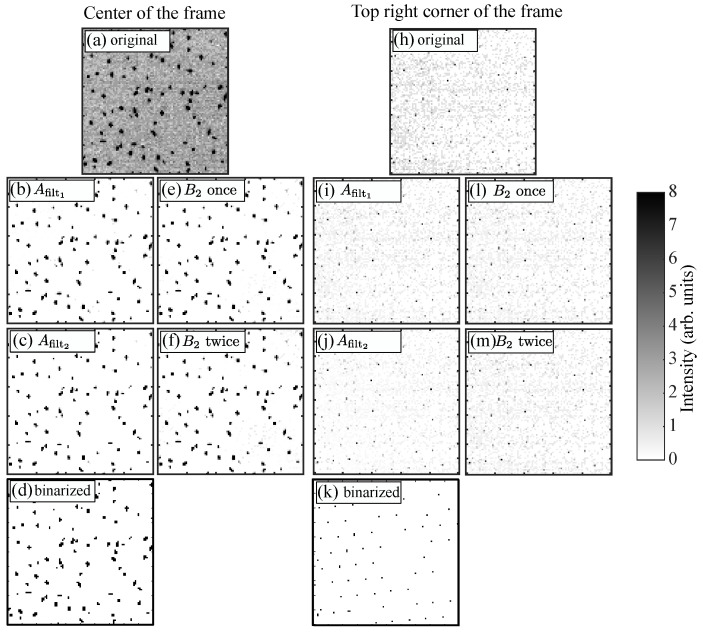
(**a**,**h**) Zoom of the raw original frame shown in Figure 2 in the centre and at the top right corner, respectively. Note that the grey scale of the original image has been changed from 0–255 to 0–8 for better visibility (pixels artificially saturate if their intensity is greater than 8). (**b**,**i**) Respective zooms of the frame after removal of the background obtained by blurring the original frame using a Gaussian filter having a width σ=50. (**c**,**j**) Respective zooms of the frame after the second-pass background removal obtained by blurring the images in (b,i) using a Gaussian filter having a width σ=10. (**d**,**k**) Binary images obtained after thresholding the images in (**c**,**j**) with a threshold value of 1.8. (**e**,**l**) Respective zooms of the frame after removal of the background obtained by blurring the original frame using a Gaussian filter having a width σ=10. (**f**,**m**) Respective zooms of the frame after the second-pass background removal obtained by blurring the images in (**e**,**l**) using a Gaussian filter having a width σ=10.

**Figure 4 jimaging-05-00041-f004:**
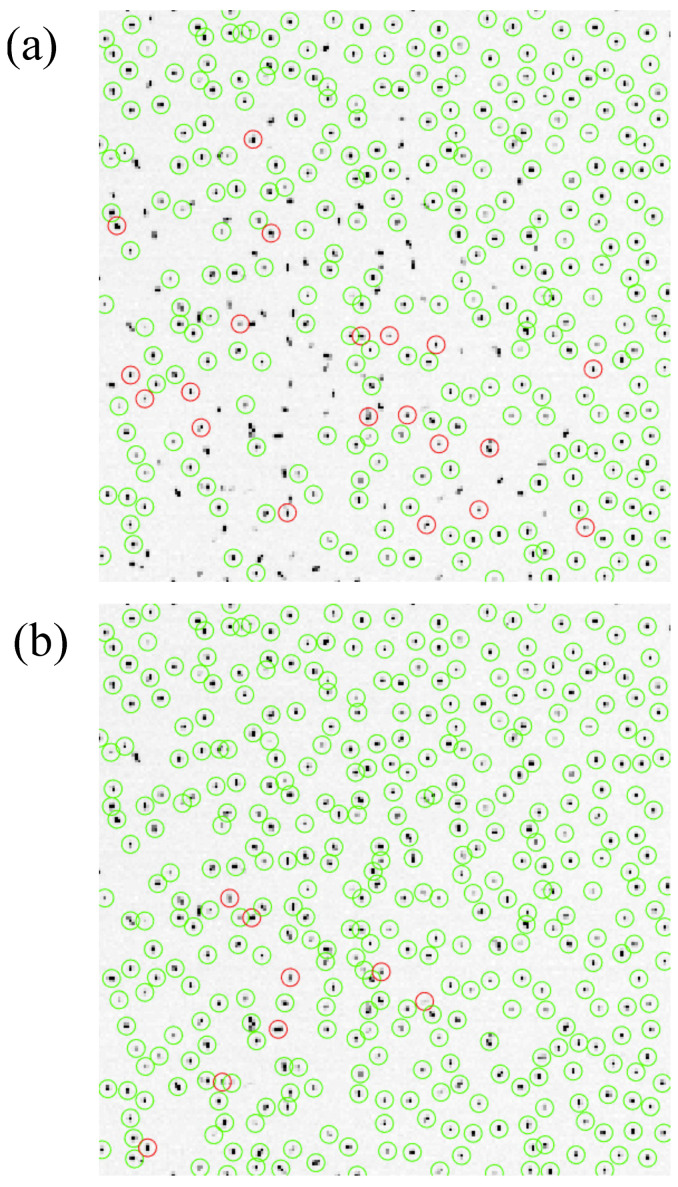
Particles detected and tracked for more than 3 frames at t=8 s. The green circles correspond to the particles successfully linked to particles detected in the previous frame. The red circles correspond to new (rediscovered) particles and tracked in a minimum of 2 frames afterward. (**a**) Using $R_{i_{MP}}\left(t_{N+1}\right)$ as starting point for locating the particles (quasi-static approximation). (**b**) Using $R_{i_{Pred}}\left(t_{N+1}\right)$ as starting point for locating the particles (position prediction). The tracking results are plotted over the original frame. Note that the grey scale of the original image has been changed from 0–255 to 0–64 for better visibility (pixels artificially saturate if their intensity is greater than 64).

**Figure 5 jimaging-05-00041-f005:**
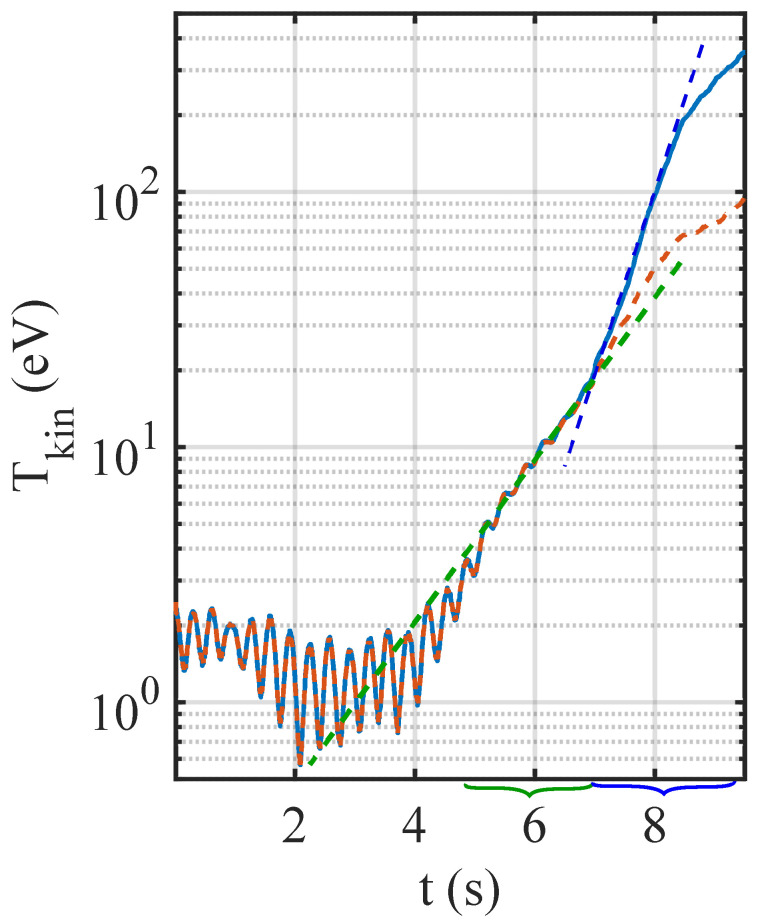
Evolution of the microparticle kinetic temperature $T_{\mathrm{kin}}$ using the two different linking strategies: Dashed orange curve: Using $R_{i_{MP}}\left(t_{N+1}\right)$ as starting point (quasi-static approximation). Plain blue curve: $R_{i_{Pred}}\left(t_{N+1}\right)$ as starting point (position prediction). The dashed green and blue lines represent the energy growth during the crystalline mode coupling instability (MCI) and fluid MCI, respectively. The green and blue curly brackets show the periods of time corresponding to the crystalline MCI and fluid MCI, respectively. Note the difference in energy growth during fluid MCI when using the different linking strategies.

**Figure 6 jimaging-05-00041-f006:**
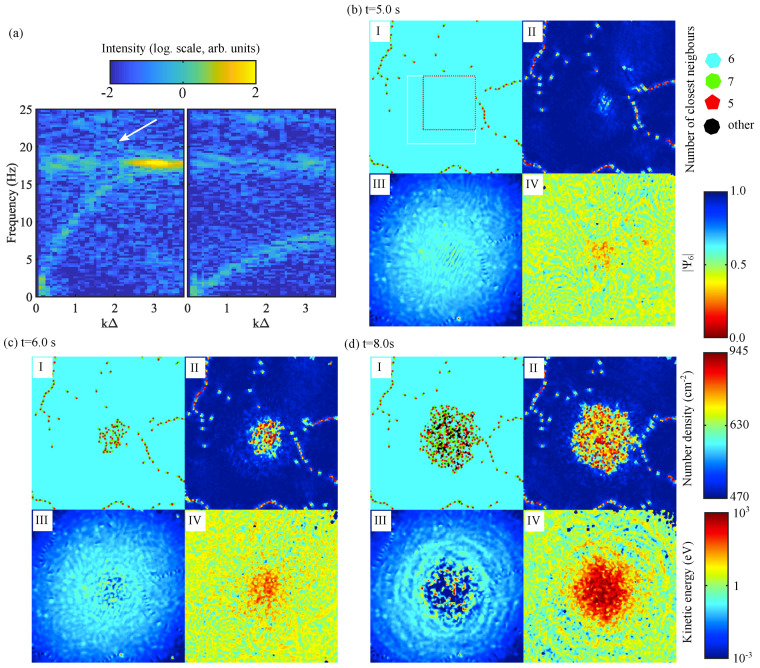
(**a**) Longitudinal (left) and transverse (right) current fluctuation spectra of the monolayer in the crystalline state for $\theta =0^{\circ }$. The arrow shows the traces of mixed polarisation. (**b**) Monolayer at t=5.0 s. (**c**) Monolayer at t=6.0 s. (**d**) Monolayer at t=8.0 s. In (**b**–**d**), I shows the map of defects. The number of closest neighbours is colour-coded; II shows the value of the bond orientational order $|\psi_{6}|$; III shows the particle number density; IV shows the particle kinetic energy (global drift is not removed). In (**b**)I, the white square shows the area used to compute the current fluctuation spectra shown in (**a**), and the red square shows the area used for computing the evolution of the kinetic temperature shown in Figure 5.

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
