# Peer review of "Tracking and Linking of Microparticle Trajectories During Mode-Coupling Induced Melting in a Two-Dimensional Complex Plasma Crystal"

_2313-433X, 2019, doi:10.3390/jimaging5030041_

Round 1
Reviewer 1 Report
In the present paper a method for tracking microparticles in a 2D plasma crystal is discussed. This method is an improvement of former methods used by the authors. Unfortunately, it is not compared to many other methods used for similar purpose (see e.g. N. Chenouard et al., "Objective comparison of particle tracking methods", Nature Methods 11 (2014) 281). Although the physical results presented in this paper are very interesting, the image analysis method used here is rather standard. Further improvements mentioned in the last paragraph of the paper could be an interesting step forward deserving a publication in J. Imaging.
Author Response
We would like to thank the referee for stating that “the physical results presented in this paper are very interesting”.
PTV techniques used in complex plasma studies are generally similar to those used for colloids and biological systems (such as those presented in Refs. [49–52]). We do thank the referee for the new reference ([52] in the revised manuscript), which have certainly helped us to better focus the aim of our article.
Indeed, as stated by Chenouard et al. [52], “there exists no universally best method for particle tracking” and “a method reported to work for certain experiments may not be the right choice for [another] application”. Therefore, as was already said, the aim of the present study is not to deliver the best method for particle tracking in quasi-2D complex plasma crystals but rather to propose an adapted method that gives results of goodquality for the detailed investigation of the MCI-induced melting.
We agree with the referee that a comparison of all the tracking algorithms would be very interesting but this is beyond the scope of the present paper and would require, as for Ref. [52], the involvement of many researchers over an extended period of time. Moreover, the improvements described in the last paragraph are currently under development and are not yet ready for publication.
However, we would like to insist on two main differences between our study and the one suggested by the referee: (1) the mean number density of our crystalline monolayer in the camera field of view was ~530 cm-2(~9800 particles in each frame of 1024x1024 pixels)which is almost twice the maximum density of Ref. [52].(2) Two types of noise usually coexist in the frame: (i) noise due to inhomogeneous illumination and (ii) random noise due to the digitization of the image in the video camera. In the presented experiment, it resulted in a strongly changing SNR(~10 at the center of the frame and ~3at the edges). We believe that for these reasons, in addition to the presentation of a method adapted to a specific problem, and as stated by the two other referees, our article deserves publication in J. Imaging.
This comment has been partially added to the manuscript.
p { margin-bottom: 0.25cm; direction: ltr; line-height: 115%; text-align: left; background: transparent none repeat scroll 0% 0%; }a:link { color: rgb(0, 0, 128); text-decoration: underline; }a.cjk:link { }a.ctl:link { }Reviewer 2 Report
The paper presents a method for tracking individually micron-size dust spheres floating in plasma in order to infer their trajectories. The technique employs a low pass filter (Gaussian in 2 D) for removing noise in the acquired images and a sorting strategy for identifying the particles. The technique is then applied to a particular experimental case by analyzing the dynamical state of a plasma crystal. Initially the crystal is stable with a hexagonal symmetry visible in the horizontal plane, and then is destabilized by the so-called MCI instability when one of the system parameters (radio-frequency power) is varied.
The results are interesting and worthwhile of publication in the Journal of Imaging as they can be of help for the dusty plasma community and for other interested experimentalists.
The paper can be published if the authors clarify some issues:
1. Fig. 1b) is a very clear and defined image of the crystal and its inverted version would be very easy to work with. It is not clear if this figure is a cut from Fig. 2 a). On the other hand Fig. 2 a) is not as well defined as fig. 1 b).
2. It would be helpful to be able to see a spatial scale in the images of Fig. 2.
3. Since the images 3 d) and 3 e) are similar, it is probably useless to show them both. It is already mentioned in the text that after the second application of the Gaussian filter there is no visible improvement.
4. In Fig. 6 the characters used for identifying the sequence of images (i.e. i, ii, iii ) should be enlarged as they hard to spot. Besides, one may be tempted to associate them with the index i used in some of the formulas.

Author Response
Thank you for stating that our article is “interesting and worthwhile of publication in the Journal of Imaging”
1. Fig. 1b) is a very clear and defined image of the crystal and its inverted version would be very easy to work with. It is not clear if this figure is a cut from Fig. 2 a). On the other hand Fig. 2 a) is not as well defined as fig. 1 b).
Images 1(b) and 1(c) have been heavily reworked (non-linear contrast and luminosity enhancements, Gaussian smoothing) and are therefore for illustration only. This comment has been added to the manuscript.
2. It would be helpful to be able to see a spatial scale in the images of Fig. 2.
A scale has been added to Fig 2a.
3. Since the images 3 d) and 3 e) are similar, it is probably useless to show them both. It is already mentioned in the text that after the second application of the Gaussian filter there is no visible improvement.
We agree with the referee that these image could be removed. However, an illustration is sometimes more efficient than a sentence and since we are not space-limited we have decided to keep them.
4. In Fig. 6 the characters used for identifying the sequence of images (i.e. i, ii, iii ) should be enlarged as they hard to spot. Besides, one may be tempted to associate them with the index i used in some of the formulas.
We have increased the size and changed to capital letters.
p { margin-bottom: 0.25cm; direction: ltr; line-height: 115%; text-align: left; background: transparent none repeat scroll 0% 0%; }a:link { color: rgb(0, 0, 128); text-decoration: underline; }a.cjk:link { }a.ctl:link { }Reviewer 3 Report
The authors provide a description of their two-pass image processing technique for enhanced particle detection and compare two different frame-to-frame linking strategies to identify the trajectories of particles moving with high velocities. The results of the image processing and linking are illustrated by analyzing the melting of a crystalline lattice due to dynamic instabilities.
I found the author’s explanation of the image processing method to be very clear and easy to follow, with enough detail to easily replicate the technique. I do have a few comments which may clarify some of the explanations.
1. The explanation of Figure 3 is clear in the text, but the figure caption is a bit difficult to follow. I would suggest re-labeling the subsequent images in the order that the image processing is applied. For example, (a) is a detail from the original raw image, (b) is the result (Afilt1) after applying filter B1, (c) is the result (Afilt2) after applying filter B2, and (d) is the final image after thresholding. Images (e,f) are the results after applying filter B2 twice, which did not provide significant improvement. (With similar relabeling for the images shown on the right for the detail of the top right corner.) It may even be useful to add labels to each row of the figure: Row 1-> reduced range for grayscale, Row 2-> Afilt1, Row3-> Afilt2, Row 4-> binary thresholding.
2. In the paragraph at the top of page 7 describing the calculation of the particle velocities, I found the phrase “differentiation matrix of the filter” to be unclear: “For this purpose, a Stavitzky-Golay filter using a second order polynomial on 3 points window was applied (resulting in a piecewise exact fit of the trajectories). Then, the first derivative of the trajectories was computed directly using the differentiation matrix of the filter and the microparticle velocities vi were obtained.”
Do you mean that the first derivative was computed by differentiation of the matrix of the filtered data? I am thinking that the matrix is really just the representation of the trajectory data, so the derivative is calculated using each successive difference of the particle positions after the filter is applied.
Finally, I have one question about the kinetic temperature data shown in Figure 5. Both of the linking strategies show that the energy growth changes slope in the midpoint of the fluid MCI. Is this due to the physics of the melting, a limitation of the crystal size or is the increase in particle velocity enough that the linking algorithms again start missing the very fastest particles?
I recommend publication of this article with very minor revisions.
Author Response
Thank you for stating that the “explanation of the image processing method to be very clear and easy to follow, with enough detail to easily replicate the technique” and recommending publication.
1. The explanation of Figure 3 is clear in the text, but the figure caption is a bit difficult to follow. I would suggest re-labeling the subsequent images in the order that the image processing is applied. For example, (a) is a detail from the original raw image, (b) is the result (Afilt1) after applying filter B1, (c) is the result (Afilt2) after applying filter B2, and (d) is the final image after thresholding. Images (e,f) are the results after applying filter B2 twice, which did not provide significant improvement. (With similar relabeling for the images shown on the right for the detail of the top right corner.) It may even be useful to add labels to each row of the figure: Row 1-> reduced range for grayscale, Row 2-> Afilt1, Row3-> Afilt2, Row 4-> binary thresholding.
The figure has been redone according to the referee’s comment.
2. In the paragraph at the top of page 7 describing the calculation of the particle velocities, I found the phrase “differentiation matrix of the filter” to be unclear: “For this purpose, a Stavitzky-Golay filter using a second order polynomial on 3 points window was applied (resulting in a piecewise exact fit ofthe trajectories). Then, the first derivative of the trajectories was computed directly using the differentiation matrix of the filter and the microparticle velocities vi were obtained.”
Do you mean that the first derivative was computed by differentiation of the matrix of the filtered data? I am thinking that the matrix is really just the representation of the trajectory data, so the derivative is calculated using each successive difference of the particle positions after the filter is applied.
We have rewritten our explanation to improve clarity as follows:
The velocity of the microparticles could be calculated. For this purpose, a Savitzky-Golay filter using a second order polynomial on 3-point window was applied (resulting in a piece-wise exact fit of the trajectories). Since in each particle trajectory the data points are equally spaced by the inverse frame rate, a single set of coefficients can be applied to each piece of a particle trajectory (the 3-point window) to calculate the first time derivative of the trajectory (the velocity) at the central point of the window. Thus, the microparticle velocities vi were obtained.
3. Finally, I have one question about the kinetic temperature data shown in Figure 5. Both of the linking strategies show that the energy growth changes slope in the midpoint of the fluid MCI. Is this due to the physics of the melting, a limitation of the crystal size or is the increase in particle velocity enough that the linking algorithms again start missing the very fastest particles?
At t=8.2s, the energy growth rate decreased due to MCI saturation [34,36,65]. However, the measured kinetic temperature is underestimated because of the lost particle tracks. This is especially true for the quasi-static approximation method which cannot track particles with Ekin>470 eV.
This comment has been added to the manuscript
p { margin-bottom: 0.25cm; direction: ltr; line-height: 115%; text-align: left; background: transparent none repeat scroll 0% 0%; }a:link { color: rgb(0, 0, 128); text-decoration: underline; }a.cjk:link { }a.ctl:link { }Round 2
Reviewer 1 Report
The arguments given in the author response are convincing and the manuscript has been improved considerably. Therefore, I recommend the publcation of this paper in its present form.
Author Response
Thank you for recommending the publication of our paper in its present form.
Reviewer 2 Report
The revised manuscript can now be published.
Author Response
Thank you for recommending publication without any further revision.
Reviewer 3 Report
The authors have addressed all of my comments. I recommend publication.
Author Response
Thank you for recommending publication without any further revision